# A Framework for Pedestrian Attribute Recognition Using Deep Learning

**Saadman Sakib** [1], **Kaushik Deb** [1], **Pranab Kumar Dhar** [1] **and Oh-Jin Kwon** [2,*]

1 Department of Computer Science and Engineering, Chittagong University of Engineering & Technology, Chattogram 4349, Bangladesh; saadmansakib110442@gmail.com (S.S.); debkaushik99@cuet.ac.bd (K.D.); pranabdhar81@cuet.ac.bd (P.K.D.)
2 Department of Electrical Engineering, Sejong University, 209 Neungdong-ro, Gwangjin-gu, Seoul 05006, Korea
* Correspondence: ojkwon@sejong.ac.kr

**Abstract:** The pedestrian attribute recognition task is becoming more popular daily because of its significant role in surveillance scenarios. As the technological advances are significantly more than before, deep learning came to the surface of computer vision. Previous works applied deep learning in different ways to recognize pedestrian attributes. The results are satisfactory, but still, there is some scope for improvement. The transfer learning technique is becoming more popular for its extraordinary performance in reducing computation cost and scarcity of data in any task. This paper proposes a framework that can work in surveillance scenarios to recognize pedestrian attributes. The mask R-CNN object detector extracts the pedestrians. Additionally, we applied transfer learning techniques on different CNN architectures, i.e., Inception ResNet v2, Xception, ResNet 101 v2, ResNet 152 v2. The main contribution of this paper is fine-tuning the ResNet 152 v2 architecture, which is performed by freezing layers, last 4, 8, 12, 14, 20, none, and all. Moreover, data balancing techniques are applied, i.e., oversampling, to resolve the class imbalance problem of the dataset and analysis of the usefulness of this technique is discussed in this paper. Our proposed framework outperforms state-of-the-art methods, and it provides 93.41% mA and 89.24% mA on the RAP v2 and PARSE100K datasets, respectively.

**Keywords:** pedestrian attribute recognition; mask R-CNN; transfer learning; ResNet 152 v2; oversampling

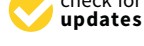

## 1. Introduction

Pedestrian attribute recognition has become popular because of its enormous application scope. It refers to recognizing the attributes of a pedestrian from a given image, i.e., gender, age, upper body dress, lower body dress, etc., as shown in Figure 1. A surveillance scenario might contain multiple pedestrians. Recognizing all the pedestrians with their attributes from a surveillance image is a very challenging task [1]. This attribute recognition task is mainly helpful for person retrieval, person re-identification, person search, etc. Machine learning and deep learning techniques are now required for automation jobs since they outperform conventional methods.

For image processing, deep learning algorithms are utilized instead of the previously prevalent handmade approaches. The primary reason for that is a deep learning-based model can automatically extract features, whereas handmade methods require user intervention. Deep neural network (DNN)-based models for image processing applications may be pretty sophisticated. There are a lot of parameters in these DNN models. Because of increases in computational power, image identification can now be made in real-time, but getting better results remains a problem. The convolutional neural network (CNN) is an essential approach for extracting characteristics from images. Deep CNN has a great learning ability due to the utilization of several function extraction stages that can automatically acquire representations from data. The availability of vast quantities of data, along

with technology breakthroughs, has spurred CNN research, and numerous promising deep CNN designs have lately been released, as proposed in Khan et al. [2].

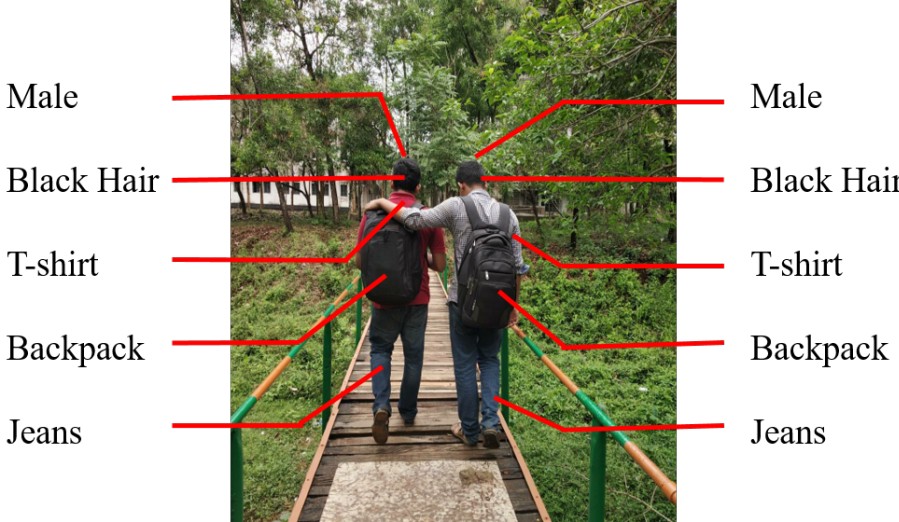

**Figure 1.** Example of the pedestrian attributes from a multiple pedestrian scenario.

This topic offers a lot of study potential because it is growing more popular and has less work. In a real-world context, it also has a wide range of applications. In a drone surveillance scenario, this may be useful. Pedestrian attribute recognition tasks have several applications, including public safety, soft bio-metrics, suspicious person identification, criminal investigation, and so on [3].

Our primary goal in this research is to recognize pedestrians in a multitude of pedestrian scenarios. We have used the Richly Annotated Pedestrian(RAP) v2 dataset to experiment. Mask RCNN is used to extract isolated pedestrians from a multiple pedestrian scenario. After that, we experimented with multiple pre-trained CNN architectures, i.e., Inception ResNet v2, Xception, ResNet 152 v2, ResNet 101 v2, to obtain a better performing architecture. In addition, an extensive experiment on the proposed CNN architecture ResNet 152 v2 is performed by fine-tuning (freezing some layers). Many previous works [4–6] applied transfer learning techniques for the pedestrian attribute recognition task. Transfer learning techniques displayed better performance in these works. In all cases, we used the transfer learning technique to obtain better results. However, the main contributions of the proposed framework are as follows:

- Proposing a framework for recognizing attributes from a multiple pedestrian scenario.
- Applying the transfer learning technique among various CNN architectures, i.e., Inception ResNet v2, Xception, ResNet 152 v2, ResNet 101 v2 to recognize pedestrian attributes.
- Tuning the best performing model ResNet 152 v2 by freezing some layers and designing a customized fully connected layer.
- Analyzing the RAP v2 dataset and applying data balancing techniques, i.e., oversampling.

## 2. Related Work

The pedestrian attribute recognition task is a series of steps. In a surveillance scenario, an image might contain pedestrians. The pedestrian attribute recognition task addresses the recognition of attributes of each of the pedestrians. The first step of the pedestrian attribute recognition task is to detect the pedestrians as accurately as possible. Several traditional and deep learning methods are applied to detect pedestrians from a surveillance scenario. After that, the next step is to recognize the pedestrian attributes. Many robust CNN architectures are used for this task.

Many traditional methods are applied to extract the pedestrians from a surveillance image. Face feature-based traditional methods are used to detect the face of a pedestrian. The use of colour space using YCbCr, HSV, etc., has a significant impact on detecting the face of multiple pedestrians [7]. Additionally, the normalization method in one of the colour space channels extended the detection accuracy. However, motion feature-based techniques are also applied for pedestrian detection. Region segmentation and machine learning-based classification technique is proposed for the pedestrian detection task [8]. Region segmentation is based on optical flow and feature extraction and classification based on Bandelet transform. Several component-based techniques are established to detect pedestrians. The main objective is to use body parts, i.e., head, arms, and leg., in an appropriate geometric configuration to detect the full pedestrian body. An AdaBoost learning algorithm is used to boost the accuracy of the Support Vector Machine(SVM) classifier for the component-based pedestrian detection task [9].

Recently, deep learning techniques have become very popular for object classification, object recognition, object detection tasks. The advancement of GPU, as well as image data, has made deep learning more efficient. Deep learning-based systems may recognize a single object or multiple objects in an image more accurately than traditional object detection algorithms. Many advanced deep learning methods are applied for the pedestrian detection task. A Convolutional Neural Network is proposed to detect the pedestrians from a surveillance scenario [10]. Three deep neural networks, i.e., supervised CNN, transfer learning-based CNN, and hierarchical extreme learning machine, are used for the task. The obtained results are very satisfactory. Additionally, many double stage CNN architectures are developed for the object detection task. These frameworks are also used to detect pedestrians. R-CNN [11], Fast R-CNN [12], Faster R-CNN [13], and Mask R-CNN [14] are the double stage CNN framework capable of detecting objects very effectively. The overall challenge for these frameworks is that they are typically slow for the large computation purpose of the two stages. This challenge is overcome by the single-stage CNN framework SSD [15], YOLO [16] as they contain only one stage which is responsible for the object detection task. However, the only challenge for single-stage detectors is detection accuracy. The Mask R-CNN can detect better in the case of object detection [17]. Thus, to better detect the pedestrians for the pedestrian attribute recognition task, we introduced Mask R-CNN in our proposed framework.

The multi-label classification issue of pedestrian attribute recognition has been widely used to retrieve and re-identify individuals. Attribute analysis is becoming more common to infer high-level semantic details. Previously conducted research used conventional methods for resolving this issue. The AdaBoost algorithm can be used to make a discriminative recognition model [18]. This algorithm resolves the problem of creating a particular handcrafted feature. Another approach is the Ensemble RankSVM method [19] that solves the scalability problem caused by current SVM-based ranking methods for the pedestrian attribute recognition task. This new method requires less memory while maintaining adequate performance.

A new re-identification technique is applied based on the mid-level semantic features of the pedestrian attributes [20,21]. The model learns attribute-centric feature representation based on the body parts. The method to determine the mid-level semantic properties are also discussed in this work.

Later, a large-scale dataset was published [22] that makes learning robust attribute detectors with strong generalization efficiency much more effortless. In addition, the benchmark output was presented using an SVM-based method, and an alternative approach was suggested that uses the background of neighbouring pedestrian images to enhance attribute inference. Additionally, a Richly Annotated Pedestrian (RAP) dataset was published [23] with long-term data collection from real multi-camera surveillance scenarios. The data samples are annotated with not only fine-grained human attributes but also environmental and contextual variables. They also looked at how various ecological and contextual variables influenced pedestrian attribute recognition.

Recently, attribute recognition is becoming more prevalent for customizing a deep learning architecture in an individual re-identification challenge. Early research focused on datasets of pedestrian attributes that were relatively thin. In video surveillance, a part-based feature representation technique is suggested [24] for human features such as facial hair, eyewear, and clothing colour. However, as opposed to current methods, the results are dismal. However, deep neural networks have shown significant improvements in the field of computer vision. Multiple CNN architecture based pedestrian attribute recognition methods, i.e., DeepSAR (with independent attributes) and DeepMAR (with dependent attributes), are also introduced [1]. They surpassed the MRFr2 system, which is state-of-the-art. A technique for simultaneously retraining a CNN architecture is proposed [25] for all attributes. It makes use of attribute interdependence, utilizing just the picture as input and no external posture, part, or context information. Their methodology is unique in its transdisciplinary nature. On two publicly available attribute datasets, CNN wins HATDB and Berkeley Attributes of People.

Likewise, a part-based system is proposed [26] in which an image is divided into 15 overlapping image parts, and each component is fed into a multi-label Convolutional Neural Network (MLCNN) at the same time, in contrast to current methods that presume attribute independence during prediction. On the Vipers and GRID datasets, experimental findings show that the model performs better. Even so, it does not demonstrate a high level of efficiency. The network AlexNet is fine-tuned [27] to make it encode an image into a discriminative function based on the corresponding attributes. The seven categories of attributes considered are gender, age, luggage-style, upper body clothing type, upper body colour, lower body colour, and lower body clothing type. The results of the experiment show that fine-tuning using attribute information enhances re-identification accuracy. However, the results are still unsatisfactory. Additionally, a grouping technique based on a fine-tuned VGG-16 structure is proposed [6]. After grouping the characteristics, they are put into a pre-trained VGG-16 model with slight performance enhancements. However, a handmade CNN model for a specific job outperforms a pre-trained model such as AlexNet, or ResNet [28].

As current approaches have difficulty localizing the areas corresponding to different attributes, a novel Localization Directed Network is proposed [29] based on the connection between previously extracted proposals, attribute positions, and attribute-specific weights to local features. An attribute aware pooling algorithm is proposed [30] that explores and exploits the association between attributes for the pedestrian attribute recognition challenge, extending the strength of deep convolutional neural networks (CNNs) to the pedestrian attribute recognition issue. Mutual learning of three attention mechanisms, i.e., Parsing attention, Label attention, and Spatial attention, is proposed [31] for pedestrian attribute analysis to select relevant and discriminative regions or pixels against variations.

Additionally, an attention-based neural network made up of a Convolutional Neural Network, Channel Attention (CAtt), and Convolutional Long Short-Term Memory (ConvLSTM) (CNN-CAtt) is also suggested [32] for the pedestrian attribute recognition task. They intended to address the issue of pedestrian attribute recognition methods already in use. The reason for the poor performance is that they ignored the relationship between pedestrian attributes and spatial information. A novel Co-Attentive Sharing (CAS) module is proposed [33] to address the same problem, which extracts discriminative channels and partial regions for more efficient feature sharing in multi-task learning.

An image-attribute reciprocal guidance representation (RGR) method is proposed [34] to explore the image-guided and attribute-guided features. They considered concrete attributes, i.e., hairstyle, shoe style, and abstract attributes, i.e., age range, role types. Moreover, a fusion attention method is used to allocate different attention levels to certain RGR aspects. Furthermore, a combination of focal and cross-entropy loss is used to solve the problem of attribute imbalance.

A novel weighted cross-entropy loss function-based work has been introduced to reduce the class imbalance problem for the pedestrian attribute recognition task [35]. GoogleNet is used as the base CNN architecture, and features from different layers of the architecture are passed into a Flexible Spatial Pyramid Pooling layer (FSPP). The outputs of these FSPP are passed to a neural network for classification purposes.

Additionally, an adaptively weighted deep framework is proposed [36] to learn the multiple person attributes. Moreover, validation loss is used to automatically update the weights of the CNN architecture, i.e., ResNet 50. Their work's main motive is to prioritize the important task by adding higher weights on those tasks.

Similar approaches, i.e., clothing attribute prediction, related to pedestrian attribute recognition, can help in the person re-identification task. A task-aware attention mechanism is proposed [37] for the clothing attribute prediction task to explore the impact of each position across different tasks. A cloth detector is used to explore the target region, and a CNN architecture extracts the features of that region. The task and spatial attention modules are used to learn the feature maps. For optimization, the t-distribution Stochastic Triplet Embedding loss function is used.

As a result, pedestrian attribute recognition has been significant for years. Additionally, deep learning models combined with a broad range of tuning and applying transfer learning techniques have recently surpassed established state-of-the-art methods. In our proposed framework, we used transfer learning techniques and tuning of the hyperparameters to obtain satisfactory results. Additionally, implementing this research in a surveillance situation is a novel notion that helps to increase people's security.

## 3. Proposed Approach

A real-world surveillance scenario contains multiple pedestrians in a single image. Thus, the first step for the pedestrian attribute recognition task is to detect the pedestrians from a multiple pedestrian scenario. Mask R-CNN is capable of object detection tasks better than other CNN architectures by 47.3% mean precision [38]. We have used the Mask RCNN object detector to isolate the pedestrians from a multiple pedestrian scenario in our proposed framework.

A similar study [39] used the R-CNN framework for clothing attribute recognition. The proposed regions of the clothes are extracted by applying the modified selective search algorithm. Then, the Inception ResNet v1 CNN architecture is used to classify the proposed regions. However, the approach is limited to only a single clothing scenario. Additionally, the Mask R-CNN does not require a selective search algorithm for the proposed regions and can perform efficiently in multiple pedestrian scenarios. Our proposed Mask R-CNN-based framework with the fine-tuned ResNet 152 v2 CNN architecture can efficiently recognize pedestrian attributes.

The next step is to recognize the attributes of each pedestrian. An image contains spatial features which are responsible for recognizing an object because of its unique characteristics. Convolutional Neural Network is the best choice to capture the spatial features responsible for recognizing an object. However, obtaining optimum performance from a CNN architecture, some prepossessing is required for the images, i.e., resizing, scaling, augmentation, normalization. In our proposed framework, we used the aforementioned preprocessing techniques. After that, we proposed a CNN architecture by experimenting with different CNN architectures to extract the spatial features. Then, the spatial features are passed to a classifier which is a customized, fully connected layer. The classifier predicts the pedestrian attributes for each pedestrian from the multiple pedestrian scenario. Figure 2 shows the proposed framework for the pedestrian attribute recognition task.

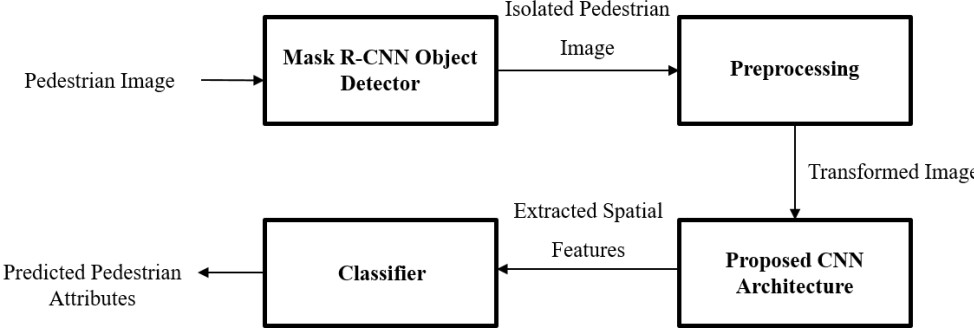

**Figure 2.** Steps of proposed pedestrian attribute recognition framework.

### 3.1. Mask R-CNN Object Detector

The first stage of the pedestrian attribute recognition framework is the Mask R-CNN object detector. The purpose of this stage is to extract the pedestrians from a surveillance scenario for further analysis. Pedestrian detection has so many applications in the real world, i.e., smart vehicles, robotics, security [40]. Mask-RCNN or Mask Region-Based Convolutional Neural Network in [41] was developed as a result of a more in-depth investigation of deep learning for pedestrian detection. The overview of the Mask RCNN object detector is shown in Figure 3. It is a double stage CNN architecture. The first stage is responsible for proposing object regions. The second stage is responsible for providing the masked regions of the objects, the bounding box of the objects, and the classification of the objects. Each stage of the Mask R-CNN is explained briefly in the following:

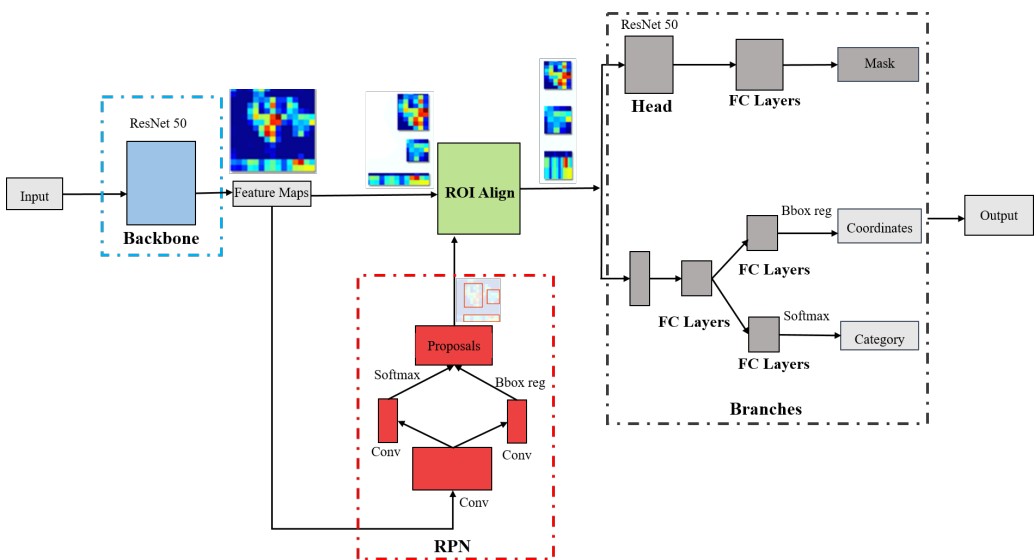

**Figure 3.** Overview of the Mask R-CNN Object Detector.

- **First Stage**

   **Backbone:** The first stage of Mask R-CNN consists of Backbone and RPN. The Backbone of the Mask R-CNN is responsible for extracting the features of a given input image. ResNet 50 CNN architecture is used in the Backbone to extract the features. The extracted features are passed to the second stage via the ROI layer. Additionally, the features are also forwarded to the RPN network.

   **RPN:** Region Proposal Network(RPN) is responsible for proposing regions of the location of an object. Several convolution layers are used to predict the probability of an object presence using the softmax function. The bounding box of the objects is also predicted in the RPN. The RPN provides the objects to the ROI align layer, passing the individual objects to the second stage.

- **Second Stage**

   **Branches:** The objects from the ROI align layers are passed to the Branches stage. This stage has three parts, i.e., Mask, Coordinates, Category. The Mask part provides the masked regions of an object. Similarly, the Coordinate part provides the object's bounding box, and the Category part provides the object's class name.

Figure 4 shows an example of Mask RCNN in real-world scenario. The image source is available in [25].

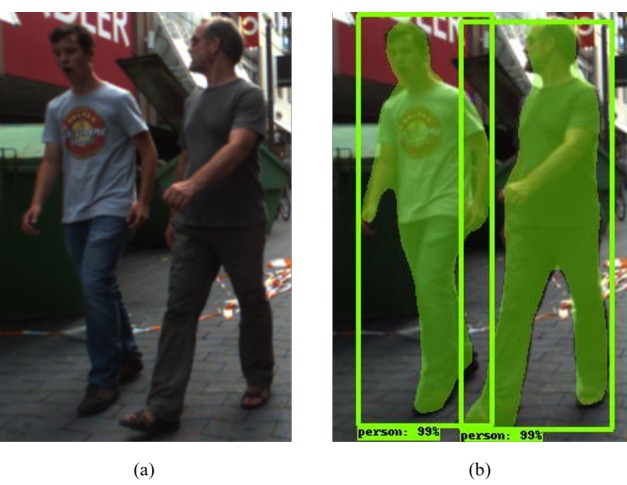

(a)                 (b)

**Figure 4.** Sample input–output of Mask RCNN object detector: (**a**) input image; (**b**) output image.

### 3.2. Preprocessing

Convolutional Neural Networks require image preprocessing for better performance. It has been proved that a prepossessed image can highlight features more than a raw image. In our proposed framework, we have applied image resizing, scaling, augmentation, and normalization techniques for capturing spatial features more robustly.

Image resizing is performed by changing the resolution of an image. As a CNN architecture can only operate on the same resolution images, we change the image resolution to $150 \times 150$. Another reason for this resolution of the images is to reduce computation cost. As the images are RGB images, the pixel values are in the range [0, 255]. To process the images faster, we divided every pixel value by 255.0, i.e., the scaling technique.

Overfitting occurs when the CNN architectures learn only on train data. As a result, it does not perform well on test data. Thus, if the variation of the image is introduced to the CNN model, then it performs better on test data. The image augmentation technique performs the variation. Our proposed framework has applied several augmentation techniques, i.e., rotation, width shift, height shift, horizontal flip, and shearing.

Normalization is applied to standardize raw input pixels. It is a technique that transforms the input pixels of an image by subtracting the batch mean and then dividing the value by the batch standard deviation. The formula for standardization is shown in Equation (1). Additionally, the effect of normalization on the performance of pedestrian attribute recognition is shown in the next section:

$$N = \frac{x_i - \mu}{\sigma} \tag{1}$$

where, $x_i$ refers to input pixel, $\mu$ refers to batch mean, and $\sigma$ refers to batch standard deviation.

When a pixel in a picture has a large number in comparison to others, and it becomes dominant. This causes the outcome of predictions to be less accurate. In this case, normalization is important as it makes a uniform distribution of the pixels and makes the pixel values smaller. It makes computation faster. In addition, normalization may cause

convergence faster than unnormalized data. A brief experiment is performed among them, which is shown in the next section.

### 3.3. Spatial Feature Extraction

The spatial features of an image are responsible for identifying an object in that image. Recent research shows that CNN architectures are very efficient in extracting spatial features. CNN architecture is a series of convolution and pooling operations. The convolution operation is performed on an image by some kernels, and it produces a feature map. After that, the feature map is passed to an activation function to introduce nonlinearity. The main purpose of convolution operation is to reduce spatial capture features as well as reduce image dimension. Then, pooling operation is performed on a given window where max or avg value from that window is taken, which is called max pool or average pool. The purpose of pooling operation is to reduce computation costs drastically. In our proposed framework, our proposed CNN architecture ResNet 152 v2 performs these similar operations in a more complex structural way. As a result, the features to recognize the pedestrian attributes are better captured.

### 3.4. Transfer Learning Approach

The concept of transfer learning arrived from the scarcity of training data. A CNN architecture, trained for a specific task on a particular dataset, can gain the knowledge of detecting edges and other low-level features. The kernels of the CNN architecture are updated during the training. These kernels are responsible for extracting features. Thus, when there is limited data for a particular task, and the goal is to better perform with this limited data, the transfer learning approach is used in this scenario. This technique uses the previously learned knowledge and applies it to a new task. Previous works [4–6] included transfer learning technique to obtain satisfactory results. We also applied the transfer learning technique on several powerful CNN architectures in our proposed framework, i.e., Inception ResNet v2, Xception, ResNet 152 v2, ResNet 101 v2. These architectures are trained on the IMAGENET dataset, which has over 14M images and 1000 classes. The best performing architecture, i.e., ResNet 152 v2, is obtained by tuning the layers of the neural network. The ResNet 152 v2 architecture has approximately 60 M parameters according to [42]. The performance comparison among the architectures is illustrated in the next section. Then, fine-tuning the ResNet 152 v2 architecture by freezing all layers except the last 4, 8, 12, 14, 20, all, and no layers is performed. The purpose is to increase the efficiency of the architecture. The details of the experiment are discussed in the next section. The architecture of the proposed ResNet 152 v2 is shown in Figure 5. A brief explanation of each of the blocks is given below:

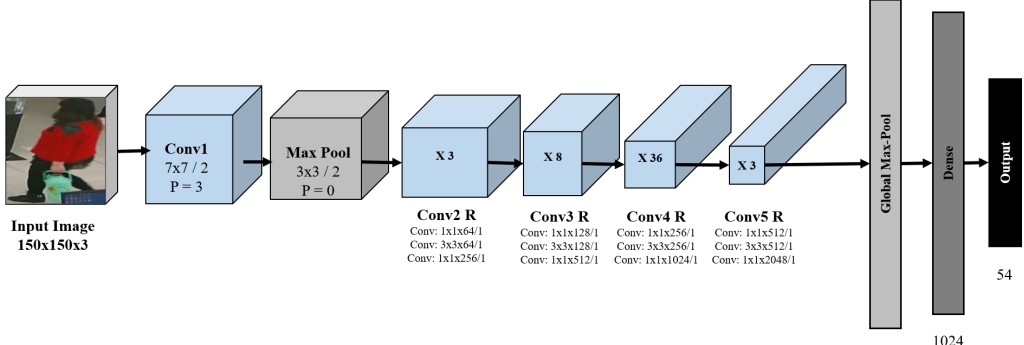

**Figure 5.** Overview of the ResNet 152 v2 architecture.

- **Conv1:** It is the first block of ResNet 152 architecture. The kernel size is $7 \times 7$ which reduces the feature map. The purpose of this block is to reduce image size.
- **Max Pool:** To extract the dominant features, initially, a max pool operation is performed. As a result, the feature map becomes smaller.

- **Conv R:** Several blocks, i.e., Conv2 R, Conv3 R, Conv4 R, Conv5 R with residual connections are shown in Figure 5. Each block has a series of convolution layers with an increasing number of channels. The convolution layer has a residual connection among them. This connection aims to make the network learn from its early layers so that the knowledge is not forgotten. The increasing number of channels indicates that the feature map gradually decreases and captures the dominant spatial features.

### 3.5. Classification

A fully connected layer consists of a series of flatten layers and dense layers. It is used for the classification of pedestrian attributes. The output layer of the fully connected layer gives values in the range [0, 1]. The sigmoid activation function is used in the output layer. The value 0 refers to the lowest probability for an attribute, and the value 1 refers to the highest probability of an attribute. The formula for the sigmoid activation function is shown in Equation (2):

$$s(z) = \frac{1}{1 + e^{-z}} \tag{2}$$

where, $z$ is the input value and $s(z)$ is sigmoid value. The input value $z$ can be calculated by using the formula shown in Equation (3).

$$z = f(w.x + b) \tag{3}$$

where, $w$, $x$ and $b$ are the weights, features and bias of the previous layer of the output layer.

After extracting the spatial features from the convolution layers, the features are passed to a classifier, i.e., a neural network. After vivid experimentation, dense layers of 1024 nodes is chosen for our ResNet 152 v2 architecture. Additionally, the output layers contain 54 nodes, as our RAP v2 dataset has 54 attributes. The details of the experiments and dataset are described in the next section.

### 3.6. Data Oversampling

As our dataset RAP v2 [43] is highly imbalanced, which is described in the next section, we also performed some data balancing experiments, i.e., data oversampling, to compare the performance of our proposed architecture in both scenarios, i.e., without oversampling, with oversampling. Some study [44] suggests that undersampling is better than oversampling on the performance on cost with the decision tree learner C4.5. In our proposed framework, we have used only oversampling technique to mitigate the class imbalance problem because the undersampling technique will greatly reduce the amount of data based on the lowest majority classes of the dataset. As the dataset contains a variety of pedestrian scenarios, reducing the variety can cause the CNN architecture to learn only from the limited variations. Thus, the overall recognition of pedestrian attributes should be limited to some scenarios. To avoid limited learning, we have used oversampling data technique in our proposed framework.

For the multi-label pedestrian attribute recognition task, we applied the oversampling technique for balancing our data. Additionally, to minimize the impact of majority class samples, we used the weighted binary cross-entropy loss function. The process to perform oversampling in our proposed framework is given below:

- Give every data a power value;
- Find the frequency of power values;
- Take the max power value;
- For each power value:

  - Get size of data = (max power value − power value);
  - Take a random copy from the dataset with the same power value;
  - Add it with the amount of size of data in the original dataset.

### 3.7. Weighted Binary Cross-Entropy Loss

This loss function is similar to the binary cross-entropy loss function. The weights of the positive and negative samples are integrated into this loss function to make the architecture learn minority samples. The following steps calculate the weights:

- For each attribute:
  - Compute positive weight, $w_p$ = Total data size/2 × Total Positive Samples.
  - Compute negative weight, $w_n$ = Total data size/2 × Total Negative. Samples
- Apply the weights to the following loss function in (4):

$$-\frac{1}{N}\sum_{i=1}^{N} w_p * y_i \cdot log(p(y_i)) + w_n * (1 - y_i) \cdot log(1 - p(y_i)) \tag{4}$$

where, $y_i$ represents the actual class and $log(p(y_i))$ is the probability of that class.

## 4. Experiments

All the experiments are performed on the Google Colab platform. Google Colab is a cloud service that provides the necessary tools for any deep learning related experiments. A total of 12 GB of RAM was used for the experiment. Additionally, for training the CNN model, we used Tesla K80 GPU with 14 GB memory. Experiments are performed in the Keras platform with Tensorflow 2.0 as the backend. The code is developed by using Python 3.7 programming language.

### 4.1. Dataset Description

There are so many publicly available datasets for pedestrian attribute recognition task, i.e., PETA [22], APiS [45], and VIPeR [46]. These datasets have some improvable areas, i.e., more variations of data collecting sources, limited annotation type, and controlled scenarios. A richly Annotated Pedestrian (RAP) dataset was proposed [23] by improving the areas as mentioned above with 41,585 pedestrian images from a real-world surveillance scenario. Additionally, the RAP dataset became more enriched by annotating more attributes, viewpoints, and occlusions for each image and adding more pedestrian images, i.e., RAP v2 Dataset [43]. Moreover, The RAP v2 dataset serves as a unified benchmark for both attribute-based and image-based person retrieval in real surveillance scenarios. The RAP v2 dataset description is shown in Table 1.

**Table 1.** RAP v2 dataset description.

| Richly Annotated Pedestrian (RAP) v2 Dataset | |
| --- | --- |
| Number of binary attributes | 69 |
| Number of multi-class attributes | 3 |
| Number of samples | 84,928 |
| Resolution ($W \times H$) | From $33 \times 81$ to $415 \times 583$ |
| Scene | Indoor |
| Number of cameras | 25 |

The RAP v2 dataset has 84,928 images of pedestrians. The dataset has varied resolution of images from $33 \times 81$ to $415 \times 583$. The sample images of the RAP v2 dataset is shown in Figure 6.

The training, testing and validation split ratio of the dataset is 6:2:2. Figure 7 shows the positive sample distribution for training, testing and validation data, respectively. Additionally, the 54 attributes are showed in these figures. From Figure 7, it is clearly visible that the dataset is imbalanced.

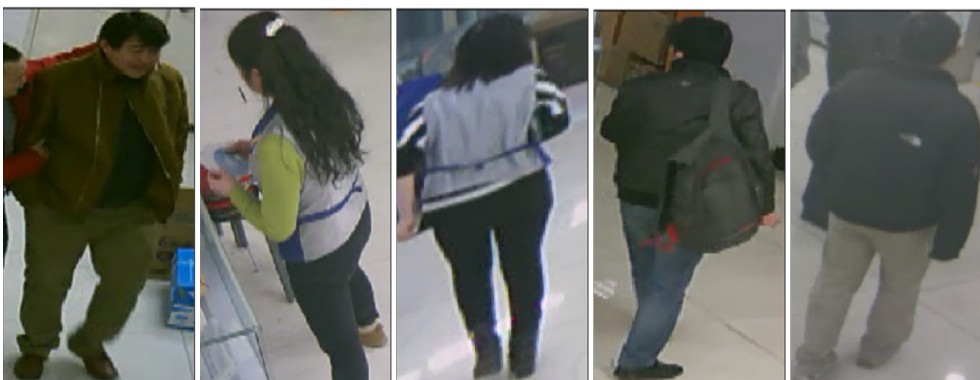

**Figure 6.** Sample images of RAP v2 dataset.

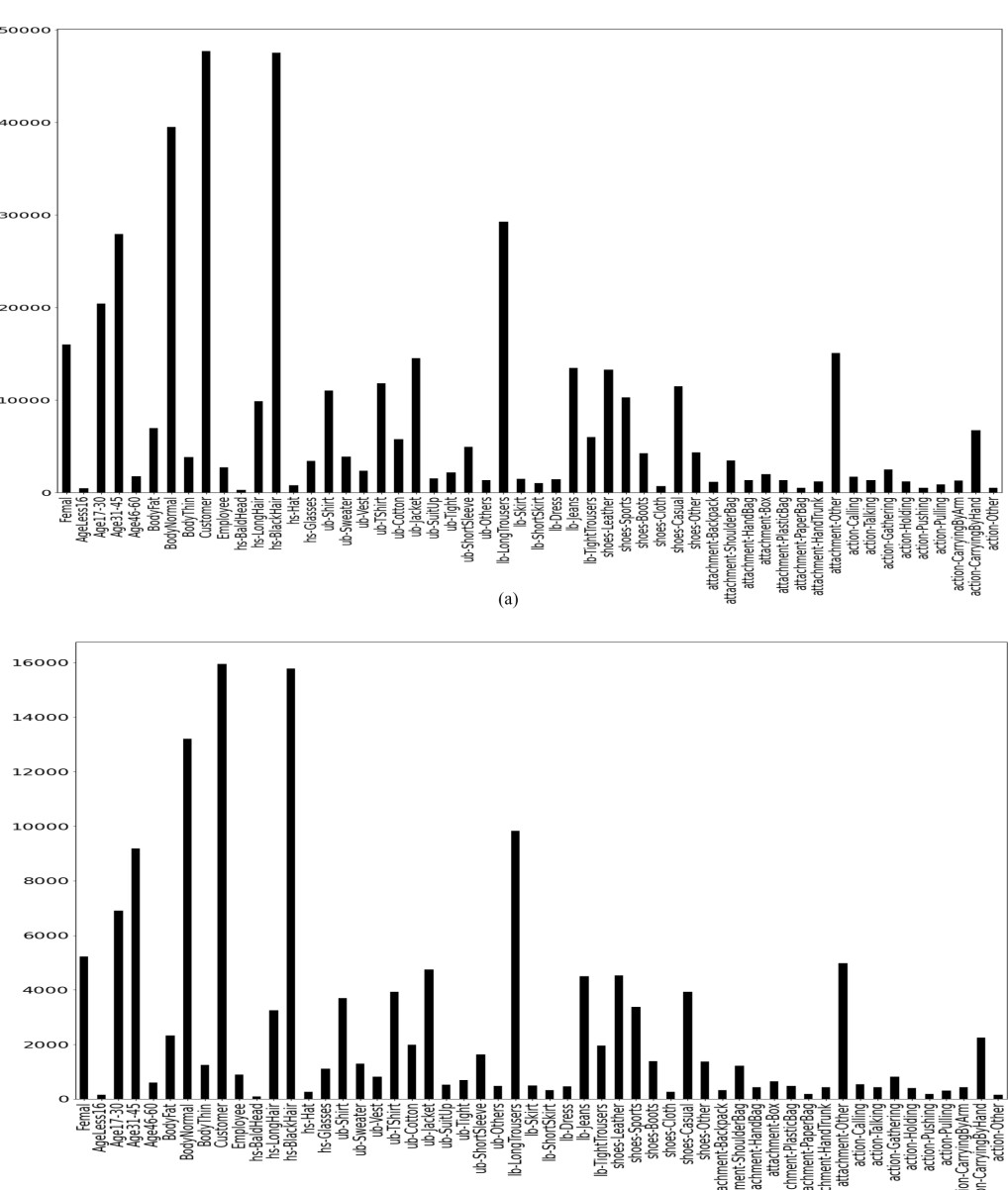

**Figure 7.** *Cont.*

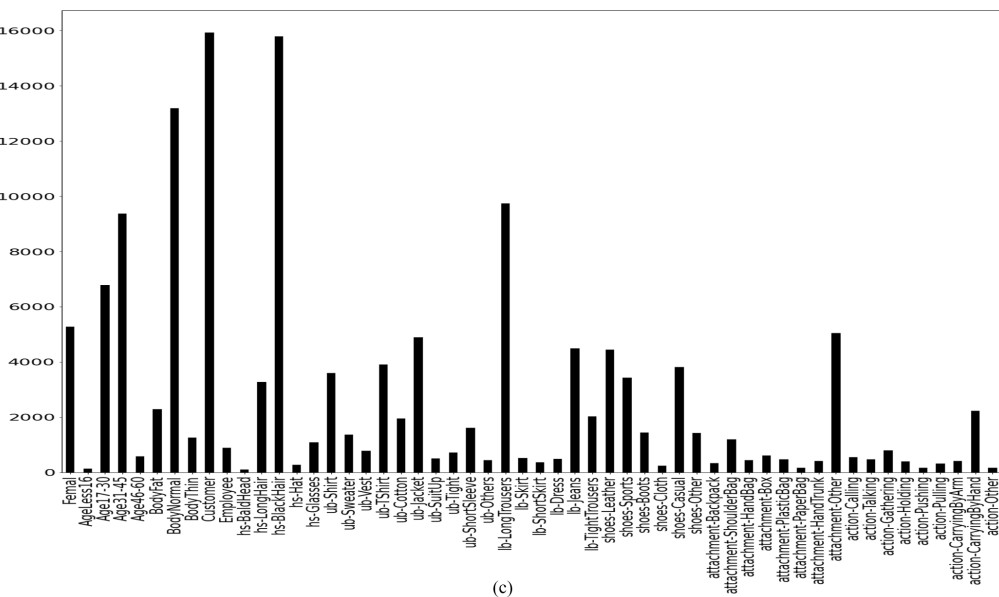

(c)

**Figure 7.** Distribution of positive samples for RAP v2 dataset: (**a**) training data; (**b**) testing data; (**c**) validation data for 54 attributes.

Additionally, we have evaluated our proposed framework on another dataset, i.e., PARSE100K [47]. The dataset contains 100,000 pedestrian images annotated with 26 attributes, i.e., gender, age, upper clothes, etc. The images are collected from real outdoor surveillance cameras. The resolution of each image is greater than $50 \times 100$. The sample image of the PARSE100K dataset is shown in Figure 8.

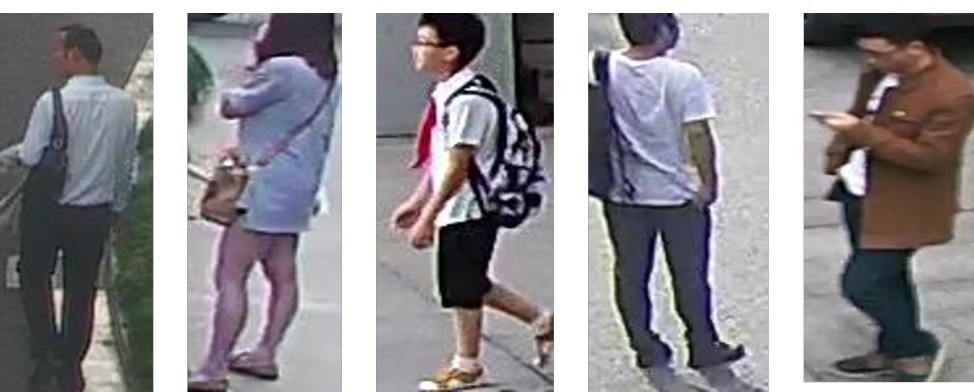

**Figure 8.** Sample images of PARSE100K dataset.

The training, testing and validation split ratio of the dataset is kept at 8:1:1 for comparing with other established methods. Figure 9 shows the positive sample distribution for training, testing and validation data, respectively, with 26 attributes.

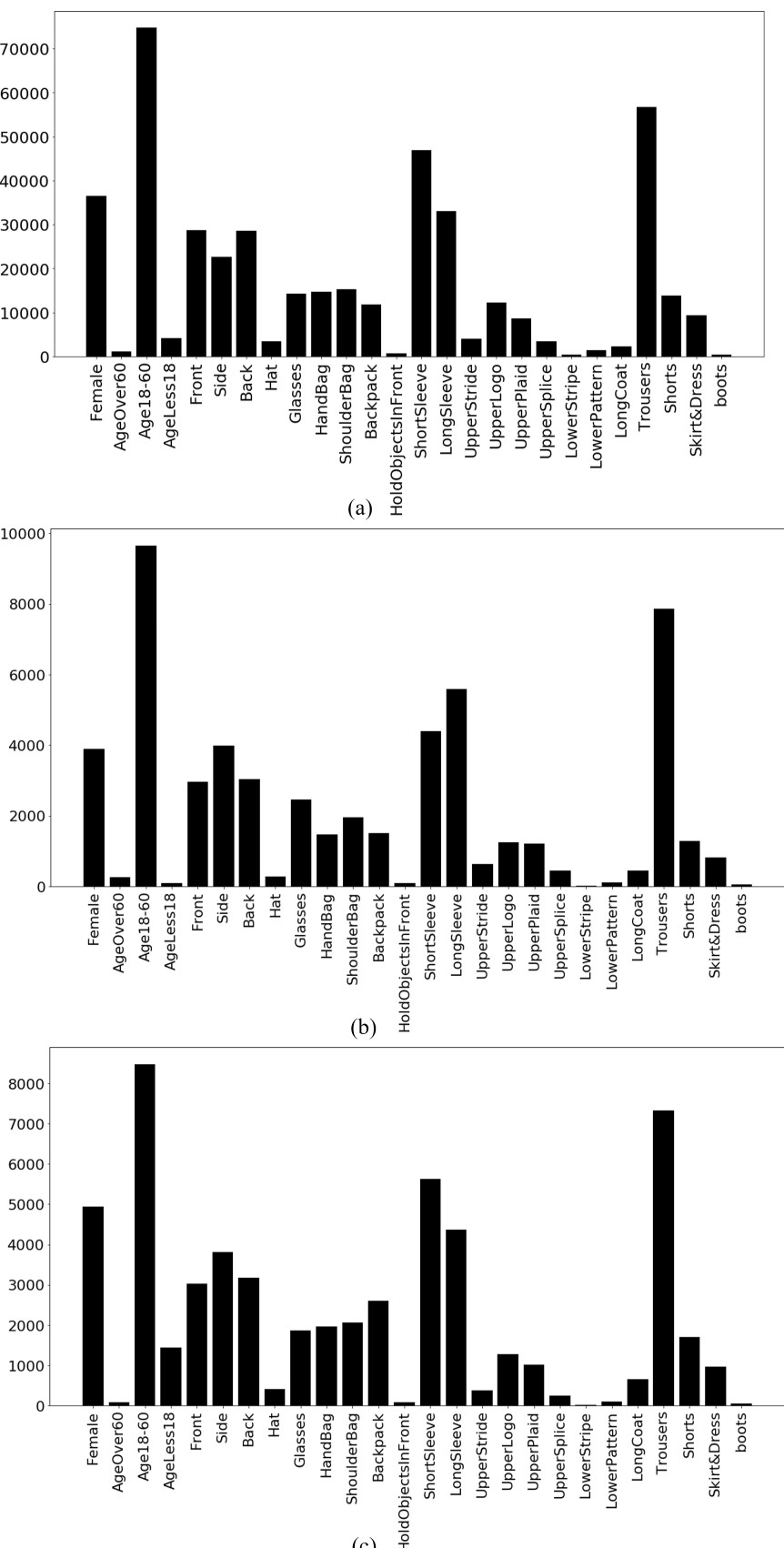

**Figure 9.** Distribution of positive samples for PARSE100K dataset: (**a**) training data; (**b**) testing data; (**c**) validation data for 26 attributes.

*4.2. Evaluation Metrics*

To assess our proposed CNN architecture, we used some metrics. Previous methods also use these metrics for evaluating their work. Accuracy, Precision, Recall, and F1 score are the most used evaluation metrics for the pedestrian attribute recognition task. These metrics help to decide which CNN architecture is more efficient and gives a better result. Multiple evaluation metrics are necessary as we can not determine the efficiency and performance of an algorithm by a single metric. As our multi-label classification task, we used a confusion matrix to visualize the per attribute result. The following equations calculate the metrics:

$$Accuracy = \frac{TP + TN}{TP + TN + FP + FN} \tag{5}$$

$$Precision = \frac{TP}{TP + FP} \tag{6}$$

$$Recall = \frac{TP}{TP + FN} \tag{7}$$

$$F1\,Score = \frac{2 \times Precision \times Recall}{Precision + Recall} \tag{8}$$

where *TP*, *TN*, *FP* and *FN* represent True Positive, True Negative, False Positive, and False Negative, respectively.

*4.3. Results and Discussion*

For experimenting with the CNN architecture for the pedestrian attribute recognition task, we split the RAP v2 dataset into training, testing, and validation data with a split ratio of 6:2:2 (50,862:16,951:16,953). There are 69 binary attributes and three multi-class attributes in the dataset. However, for comparing our work with previous works [1,25,48,49], we experimented with 54 binary attributes as in other works. Additionally, the split ratio is chosen for maintaining consistency with previous results.

The image dimension is chosen $150 \times 150 \times 3$ for our proposed framework. The reason behind it is to reduce the computation cost as larger image size increases the parameters of the CNN architecture. Hence, it also increases the computation cost. Additionally, to avoid overfitting the CNN architecture and making a better generalization for the pedestrian attribute recognition task, we applied different augmentation techniques, i.e., horizontal shift, vertical shift, rotation, shearing, and horizontal flip. Examples of these different augmentation techniques are shown in Figure 10.

Many robust CNN architectures are developed which can be used for different computer vision tasks. We experimented with powerful CNN architectures for the pedestrian attributes recognition task, i.e., Inception ResNet v2, Xception, ResNet 152 v2, ResNet 101 v2. For maintaining consistency, the experiments on these architectures are performed with the same hyperparameters. All the layers are frozen for applying the transfer learning technique. The selection of the hyperparameters is discussed further in this section. The comparison of the pre-trained models on the RAP v2 dataset is shown in Table 2. The table shows that ResNet 152 v2 CNN architecture performed better than other CNN architectures for recognizing the pedestrian attributes.

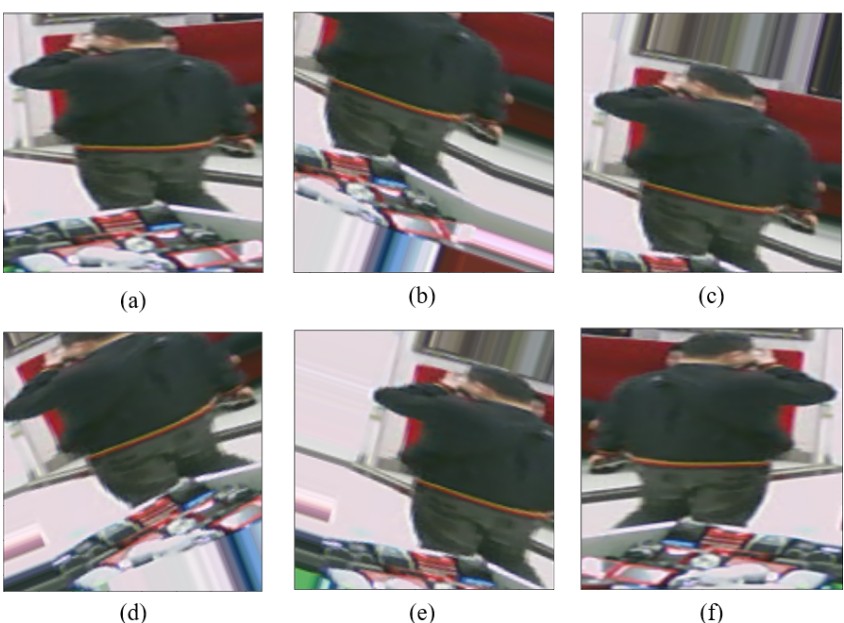

**Figure 10.** Example of augmentation techniques: (**a**) original image; (**b**) width shift by 20%; (**c**) height shift by 20%; (**d**) rotation by 25 degree; (**e**) shearing by 20%; (**f**) horizontal flip.

**Table 2.** Comparison of the pretrained models on the RAP v2 Dataset.

| Pre-trained Models | Trainable Parameters | Nodes in Dense Layer | Test Accuracy (%) |
|---|---|---|---|
| Xception | 2.1 M | | 91.56 |
| Inception ResNet V2 | 1.6 M | 1024 | 91.61 |
| ResNet 101 V2 | 2.1 M | | 92.11 |
| **ResNet 152 V2** | 2.1 M | | **92.14** |

After selecting the ResNet 152 v2 architecture as our primary architecture, we performed a vivid experiment to improve the results. A transfer learning technique requires freezing layers, which means that the weights of the frozen layers will not be updated while training the CNN architecture. In our proposed framework, we also froze some layers and experimented with them. The experimented results are shown in Table 3. The results show that Model 5 performed best among other experiments with 93.41% mA. Model 5 is obtained by freezing all layers except the last 14 layers (excluding the fully connected layers) of ResNet 152 v2 CNN architecture. The hyperparameter selection is the same as before. We used a dense layer of 1024 nodes in all cases and 40 epochs per experiment for consistency.

**Table 3.** Experiment on ResNet 152 v2 architecture on the RAP v2 dataset via the transfer learning method.

| ResNet 152 v2 Model | Trainable Layers (Excluding FC Layer) | Nodes in Dense Layer | Trainable Parameters | mA (%) | F1 (%) |
|---|---|---|---|---|---|
| Model 1 | None | | 2,184,274 | 92.14 | 27.58 |
| Model 2 | Last 4 layers | | 3,238,994 | 91.96 | 24.61 |
| Model 3 | Last 8 layers | | 3,415,040 | 92.61 | 34.08 |
| Model 4 | Last 12 layers | 1024 | 4,464,640 | 93.09 | 40.16 |
| **Model 5** | **Last 14 layers** | | **6,653,010** | **93.41** | **45.18** |
| Model 6 | Last 20 layers | | 7,879,680 | 93.01 | 37.09 |
| Model 7 | All | | 60,372,178 | 93.16 | 36.70 |

Additionally, we also experimented on the effect of normalization on our proposed framework. The performance comparison of the ResNet 152 V2 architecture on normalized and unnormalized data are represented in Table 4.

**Table 4.** Performance analysis of normalization on Model 5 on the RAP v2 dataset.

|  | Training Accuracy (%) | Validation Accuracy (%) | Test Accuracy (%) | Total Epochs |
|---|---|---|---|---|
| Normalized Data | 92.45 | 92.48 | 92.45 | 40 |
| **Unnormalized Data** | 94.25 | 93.46 | **93.41** | 40 |

Table 4 shows that our normalized data did not perform better than unnormalized data. The reason behind it is that we have already scaled our image data in the range [0, 1]. Thus, the task of normalization is already is done by scaling. This is why the normalization of the image data did not have much effect. In addition, scaling made the convergence faster in our proposed architecture as we used a larger learning rate.

In the previous section, we discussed the data oversampling technique as well as weighted binary cross-entropy loss. Data oversampling focuses on the minority class by balancing the samples with the majority classes. Figure 7 shows that the RAP v2 dataset is imbalanced due to less data of some classes, i.e., AgeLess16, hs-Baldhead, ls-skirt, attachment-Handbag, etc. After applying the data oversampling technique described in the previous section, the number of samples for the minority class becomes equal to the majority classes. The power value distribution before applying data oversampling and after applying the data oversampling technique is shown in Figures 11 and 12, respectively. The experiment details of applying data oversampling technique and weighted loss are shown in Table 5.

Table 5 shows that the experiment without oversampling performs better with 93.41% accuracy. The experiment with data oversampling increases the number of samples significantly, i.e., 462,046 images. This result proves that Model 5 performed better without the need for oversampling and weighted loss. The hyperparameters remained the same for experiment consistency.

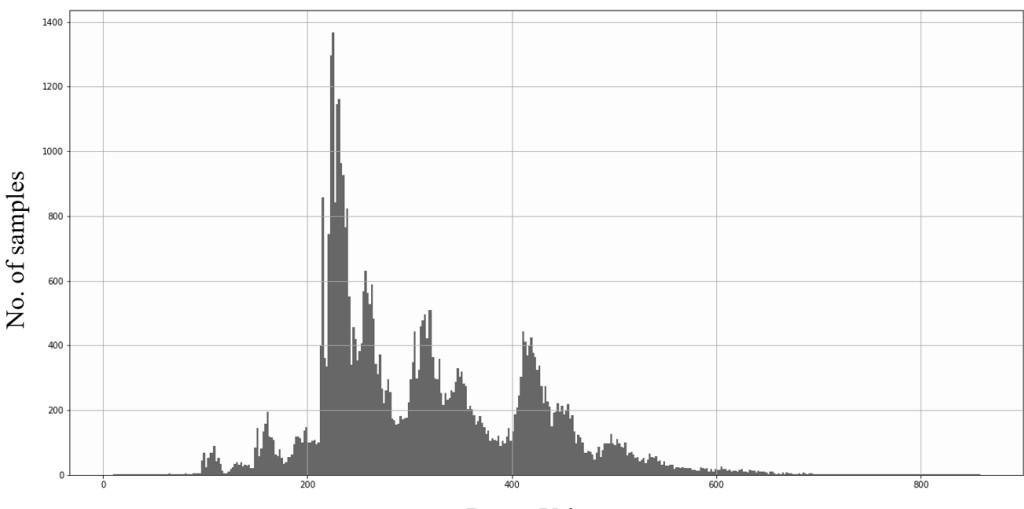

**Figure 11.** Distribution of power values before applying oversampling.

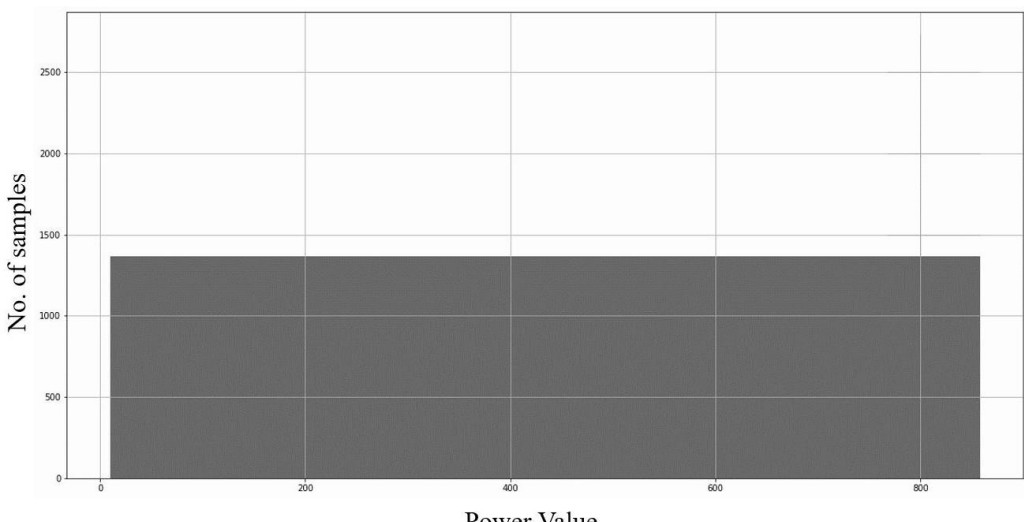

**Figure 12.** Distribution of power values after applying oversampling.

**Table 5.** Performance analysis of oversampling and weighted loss on Model 5 on the RAP v2 dataset.

| Architecture | Experiment | No. of Images | mA (%) | mP (%) | mR (%) | F1 (%) |
|---|---|---|---|---|---|---|
| Model 5 | With oversampling and weighted loss | 462,046 | 86.50 | 37.89 | 58.66 | 44.48 |
| | **Without oversampling and weighted loss** | 50,862 | **93.41** | 61.34 | 39.15 | 45.18 |

Hyperparameters are the variables whose values are not learned by the CNN architecture. Vivid experiments select these hyperparameter values. The necessity of an optimum hyperparameter for training a CNN architecture is significant. The optimum hyperparameters can reduce the training time, reduce loss function value, and increase accuracy. Table 6 shows experimentation on the different hyperparameters. Table 6 is obtained after thorough experimentation. The initial learning rate is 0.01, which will decrease by monitoring some factors, i.e., patience 3, factor 0.5, minimum delta $1 \times 10^{-4}$, minimum learning rate $1 \times 10^{-8}$. The total number of hidden layers is 1, with 1024 nodes. The batch size for training is 64. The number of epochs is 40. Additionally, the "Adam" optimizer is used for the backpropagation part of training a CNN architecture. In all experiments, these hyperparameters are selected for consistency.

**Table 6.** Hyperparameter selection from the hyperparameter space for the experiments.

| Hyperparameters | Hyperparameter Space | Optimum Hyperparameter |
|---|---|---|
| No. of Dense Layer | 1, 2, 3 | **1** |
| No. of Neurons in Dense Layer | 128, 256, 512, 1024, 2048 | **1024** |
| Learning Rate | 0.1, 0.5, 0.01, 0.05, 0.001, 0.005 | **0.01** |
| Batch Size | 32, 64, 128, 256 | **64** |
| Epochs | 10, 20, 30, 40, 50 | **40** |
| Optimizer | 'SGD', 'RMSProp', 'Adam' | **'Adam'** |

Figure 13 shows the loss and accuracy curve of Model 5 (ResNet 152 v2 architecture with trainable last 14 layers) on the datasets RAP v2 and PARSE100K. The curve shows that the loss is converged when the 40th epoch is reached. Additionally, Figures 14 and 15 show the ROC curve of Model 5 on the RAP v2 dataset and the PARSE100K dataset, respectively. The ROC curve is a graphical representation of the false-positive rate over the true-positive rate.

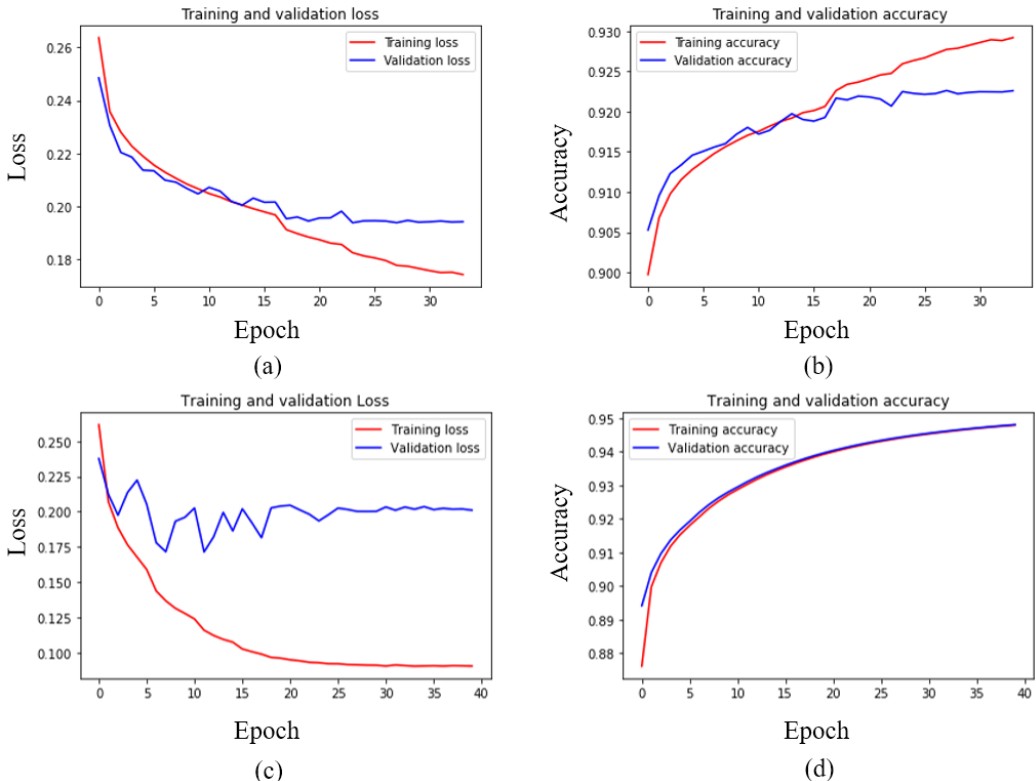

**Figure 13.** Performance of Model 5: (**a**) loss curve of the RAP v2 dataset; (**b**) accuracy curve of the RAP v2 dataset; (**c**) loss curve of the PARSE100K dataset; (**d**) accuracy curve of the PARSE100K dataset.

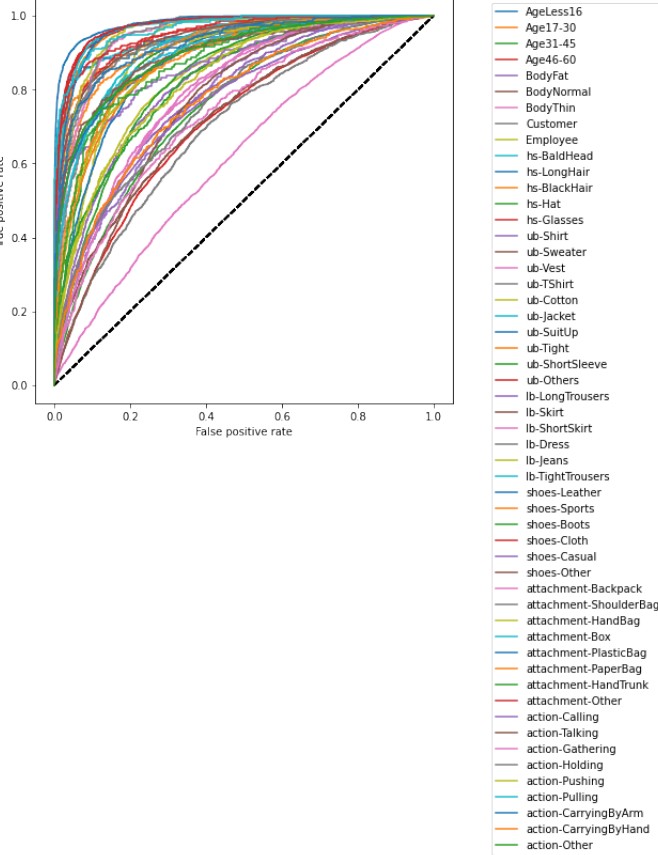

**Figure 14.** ROC curve of Model 5 on the RAP v2 dataset.

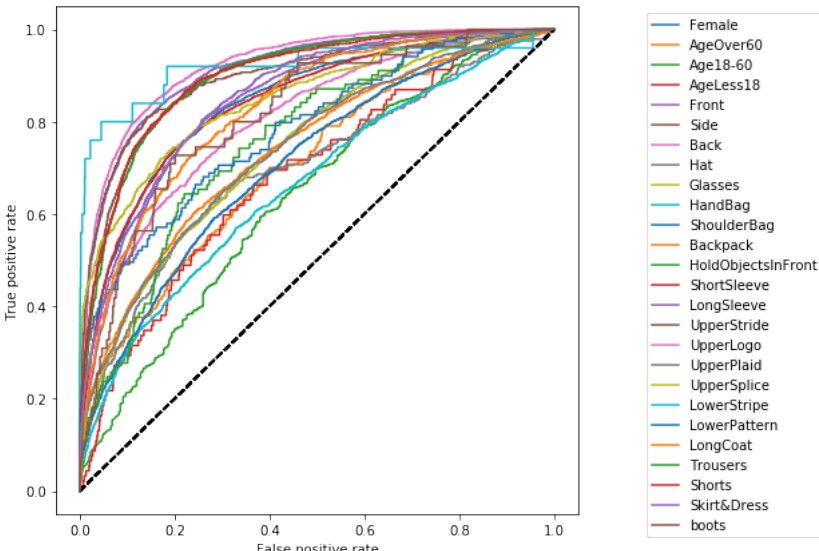

**Figure 15.** ROC curve of Model 5 on the PARSE100K dataset.

Pedestrian attribute recognition is a multi-label classification task. In general cases, a single confusion matrix is needed to represent the multi-class classification related tasks. For a multi-label classification task, a confusion matrix is required per label. Figures 16 and 17 show the confusion matrix of Model 5 on the RAP v2 dataset and the PARSE100K dataset, respectively.

The feature map developed after each block of Model 5 is shown in Figure 18. The feature map represents how an image is transformed in each layer of a CNN. The deeper the layer goes, the more complex the feature map becomes. In the last few layers, the features become uninterpretable for humans. The feature extraction process of a CNN can be compared to a black box [50]. From Figure 18, we can deduct that the performance of our proposed framework is also related to the black box concept of a CNN architecture.

Our proposed framework performs better in terms of accuracy. Tables 7 and 8 show a comparison among our proposed framework and state-of-the-art(SOTA) methods on the RAP v2 dataset and the PARSE100K dataset, respectively. Our proposed framework outperforms existing SOTA methods with 93.41% accuracy on the RAP v2 dataset and with 89.24% accuracy on the PARSE100K dataset. However, the precision, recall, and f1 score is not very high. As both datasets, i.e., RAP v2 and PARSE100K, contain a class imbalance problem, some pedestrian attributes have only a limited number of positive samples, and some attributes have a limited number of negative samples. Thus, misclassification in a class containing a limited number of positive samples can cause a major reduction in precision and recall values, which also affects the f1 score. Additionally, the different splits can have an impact on the outcome because a split can make the distribution of positive samples limited. The experiments conducted on our proposed framework is affected by the class imbalance problems described above, and hence the precision, recall and f1 score are not very high. In our proposed framework, we applied hard sharing of features. The overall extracted features are considered for the decision-making. The interdependency among the features of different pedestrian attributes is considered in our proposed framework. This factor is one of the reasons for the satisfactory performance. Our proposed framework has shown relatively small value in other evaluation metrics. The selected train, test, and validation data are different in different experiments; our selected split data are also different. Hence, the distribution of the samples is also imbalanced. After applying the data balancing techniques, the results did not improve. As a result, our proposed framework performs well in the majority of class samples. Figure 19 shows the demonstration of our proposed framework. The source of the image is available in [25].

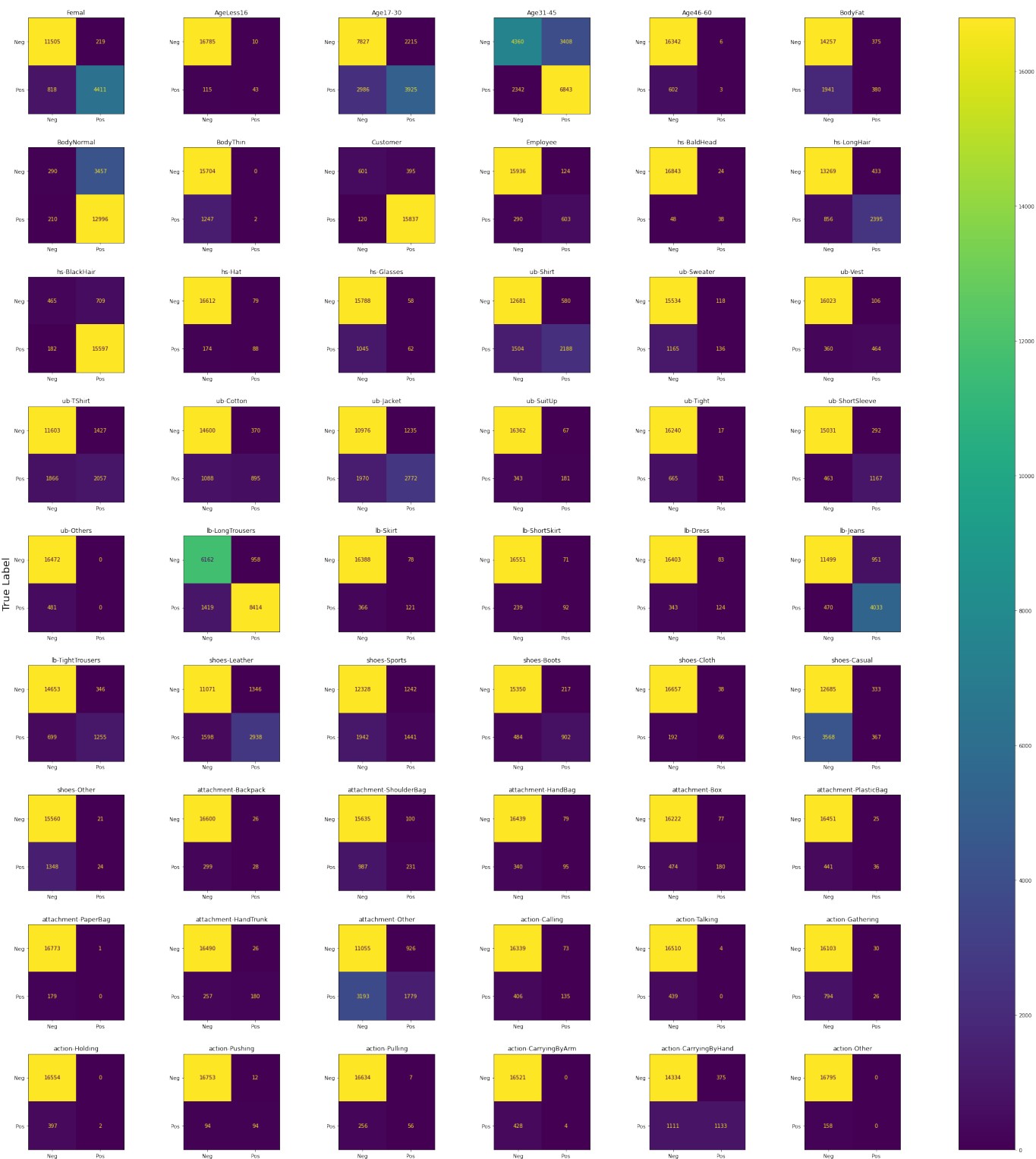

**Figure 16.** Confusion matrix of Model 5 on the RAP v2 dataset.

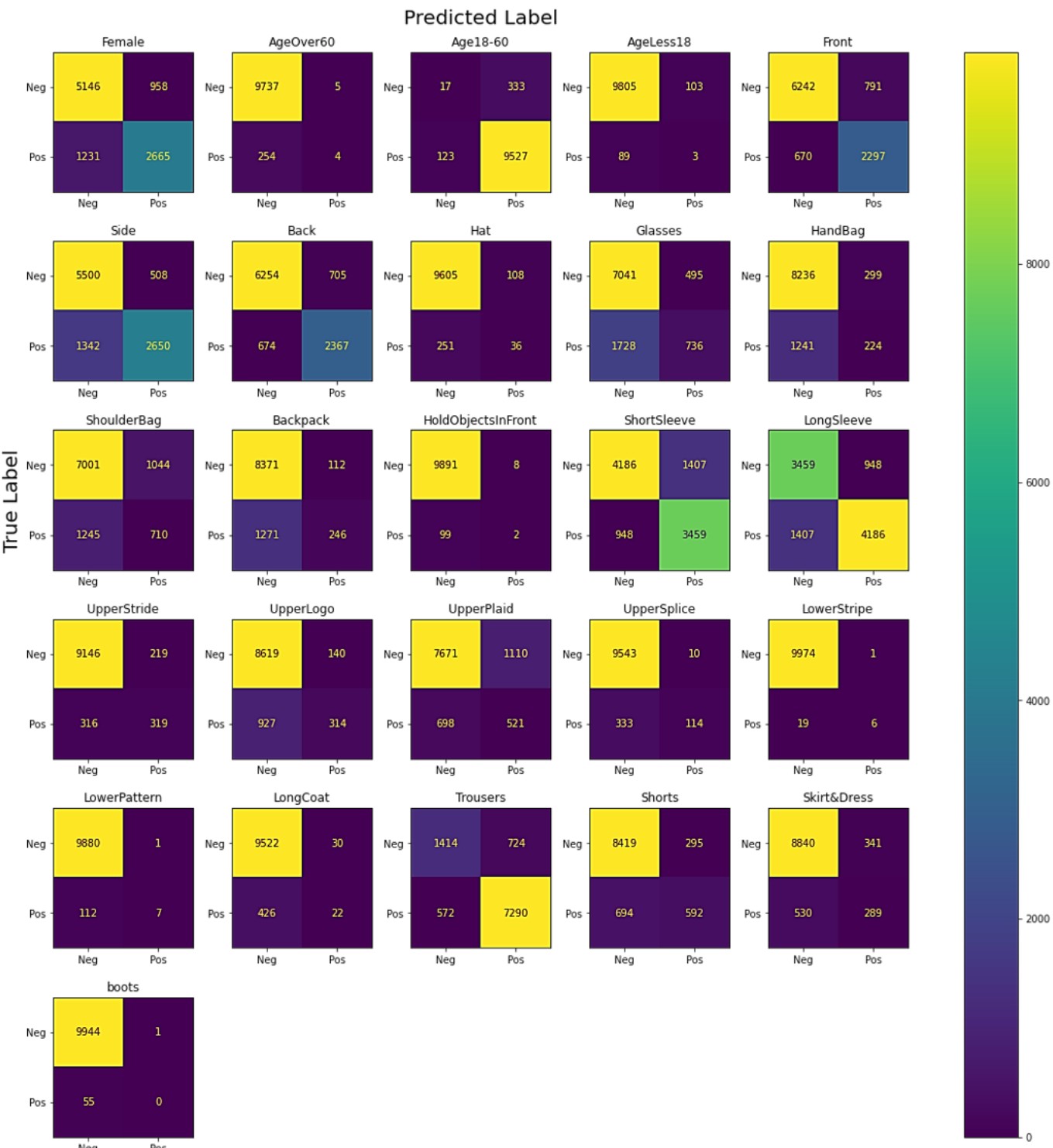

**Figure 17.** Confusion matrix of Model 5 on the PARSE100K dataset.

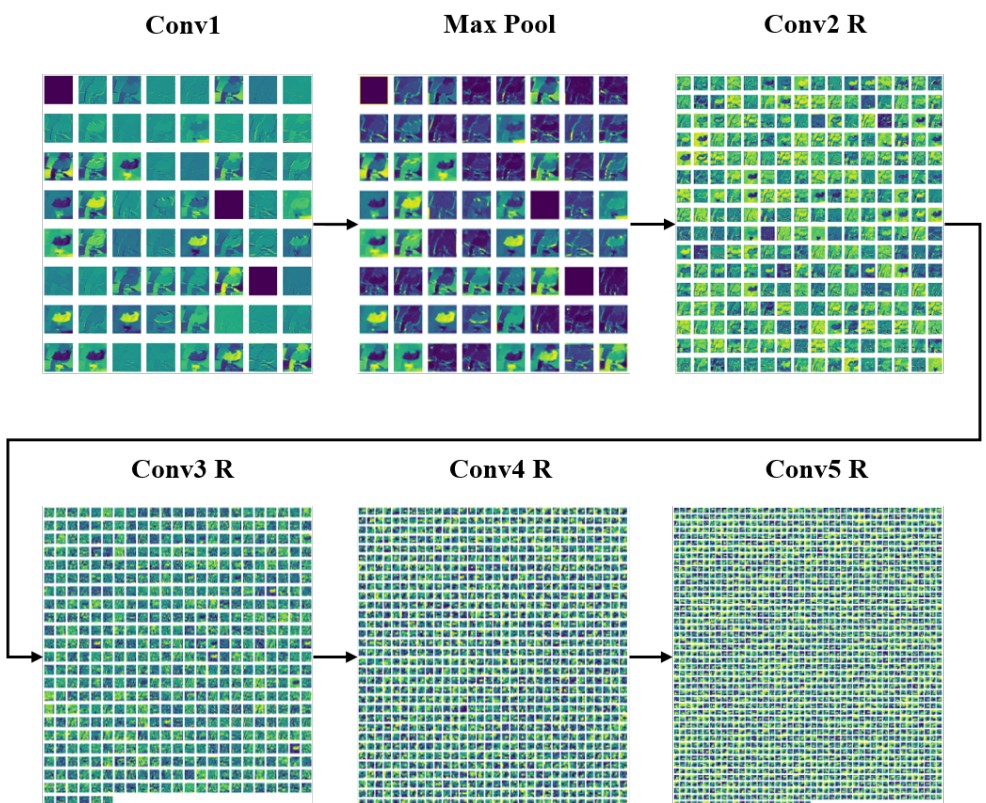

**Figure 18.** Feature map of each block of Model 5.

**Table 7.** Comparison with the state-of-the-art methods on the RAP v2 dataset.

| Method | RAP V2 Dataset | | | |
|---|---|---|---|---|
| | mA | mP | mR | F1 |
| [25] | 68.92 | 70.89 | 80.90 | 75.56 |
| [1] | 75.54 | 76.56 | 78.64 | 77.59 |
| [48] | 77.87 | 79.03 | 79.79 | 79.04 |
| [49] | 76.74 | 80.42 | 78.78 | 79.24 |
| Ours | **93.41** | 61.34 | 39.15 | 45.18 |

**Table 8.** Comparison with the state-of-the-art methods on the PARS100K dataset.

| Method | PARSE100K Dataset | | | |
|---|---|---|---|---|
| | mA | mP | mR | F1 |
| [1] | 72.70 | 82.24 | 80.42 | 81.32 |
| [47] | 74.21 | 82.97 | 82.09 | 82.53 |
| [3] | 81.63 | 84.27 | 89.02 | 86.58 |
| [33] | 77.20 | 88.46 | 84.86 | 86.62 |
| Ours | **89.24** | 58.98 | 40.02 | 42.9 |

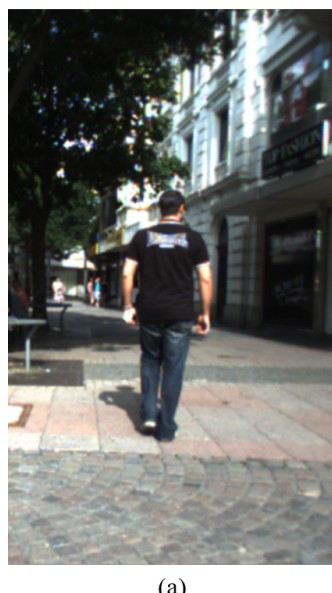 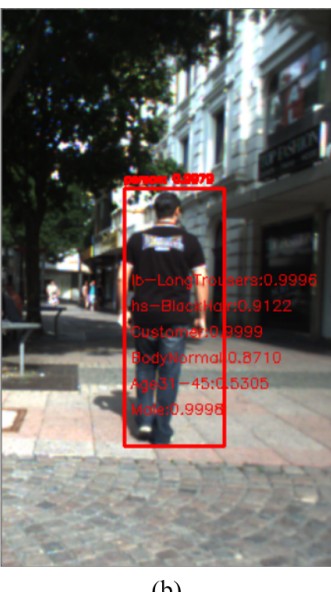

(a) (b)

**Figure 19.** Demonstration of our proposed framework: (**a**) input image; (**b**) output image.

## 5. Conclusions

In our proposed framework, we have applied the Mask R-CNN object detector to isolate the pedestrians from a multiple pedestrian scenario. Initially, ResNet 152 v2 architecture obtained better results on the RAP v2 dataset, i.e., 92.14% mA, among other architectures, i.e., Xception, Inception ResNet v2, ResNet 101 v2. Additionally, we have also proposed a fine-tuned ResNet 152 v2 (Model 5) architecture after thorough experimentation. Our proposed CNN architecture, i.e., Model 5, obtained 93.41% accuracy. Moreover, we experimented with hyperparameters to obtain a set of hyperparameters, i.e., initial learning rate 0.01, 1 hidden layer with 1024 nodes, batch size 64, number of epochs 40, and "Adam" optimizer that provided satisfactory results. Furthermore, we have displayed the dataset distribution to show the class imbalance problem. However, after applying the oversampling and weighted loss technique, the class imbalance problem is not resolved. Analysis of applying oversampling and weighted loss on Model 5 shows that our proposed fine-tuned Model 5 obtains 93.41% mA without it. Additionally, we have experimented on an outdoor-based dataset, i.e., PARSE100K, to evaluate our proposed framework in terms of generalization. Our proposed framework outperformed existing SOTA methods with 89.24% mA. The loss and accuracy curve displays the smooth training of the proposed CNN architecture. The confusion matrix and ROC show the performance of Model 5. The feature map visualization of Model 5 gives an abstract of image transformation through every block of Model 5. Our proposed CNN architecture has higher accuracy but additionally has large network parameters, i.e., 6.65 M. In the future, we tend to work with a more balanced dataset to overcome the class imbalance problem. Nevertheless, pedestrian attribute recognition has so many applications. Person re-identification is very useful in surveillance scenarios. We tend to extend our proposed framework for person re-identification tasks.

**Author Contributions:** Conceptualization, K.D.; Data curation, S.S.; Formal analysis, S.S.; Investigation, S.S.; Methodology, S.S.; Software, S.S.; Supervision, K.D.; Validation, S.S.; Visualization, S.S.; Writing—original draft, S.S.; Writing—review and editing, K.D., P.K.D., and O.-J.K. All authors have read and agreed to the published version of the manuscript.

**Funding:** This research received no external funding.

**Institutional Review Board Statement:** Not applicable.

**Informed Consent Statement:** Not applicable.

**Data Availability Statement:** The authors have used a publicly archived RAP v2 dataset, PARSE100K dataset, and PARSE-27k dataset for validating the experiment. The RAP v2 dataset is available in [43]. The PARSE-27k dataset is available in [47]. The PARSE-27k dataset is available in [25].

**Conflicts of Interest:** The authors declare no conflict of interest.

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
