# Peer review of "A Framework for Pedestrian Attribute Recognition Using Deep Learning"

_applsci, doi:10.3390/app12020622_

Round 1
Reviewer 1 Report
In this study (A Framework for Pedestrian Attribute Recognition using Deep Learning), the authors proposed a deep learning framework with mask R-CNN and different CNN architecture for attribute recognition. 80% of the total observations were used for training and validation. The rest 20% of the observations were used for testing. Transfer learning and oversampling techniques were applied to further improve the robustness of the proposed methods in the given unbalanced label dataset. Based on the experiment, the authors found that their proposed deep learning pipeline achieves the best attribute recognition performances compared to other state-of-the-art (SOTA) methods.
Overall, the manuscript is well written and decently organized. However, the proposed methods did not compare to other commonly used methods for the imbalanced datasets, such as under-sampling methods. In addition, the given proposed method was only evaluated on a single public dataset. Without evaluating the given methods on other commonly used public datasets, it is hard to believe that the given method could achieve better prediction performance compared to those currently available methods.
Q1: Similar study using the mask R-CNN with the CNN-based method has been proposed for attribute recognition [1].
Can the author explain the novelty of their own research work?
Q2: The proposed methods did not compare to other commonly used methods for the imbalanced datasets, such as the under-sampling method [2], for further evaluation. It will be better if the authors could compare other resampling methods for further comparison.
Q3: The proposed method was only evaluated on a single public dataset. It will be better if the authors could consider evaluating their proposed method in multiple publicly available datasets for attribute recognition.
Q4: Based on the testing performances in Table.7, it is very clear that the given proposed method achieved much lower prediction performances in mP, mR, and F1 values compare to other SOTA methods. Could the authors explain the reason why their method has much lower mP, mR, and F1 performances (e.g: 61.34 vs. 80.42 in mP)?
Reference:
[1] Xiang, Jun, et al. "Clothing attribute recognition based on RCNN framework using L-Softmax loss." IEEE Access 8 (2020): 48299-48313.
[2] Drummond, Chris, and Robert C. Holte. "C4. 5, class imbalance, and cost sensitivity: why under-sampling beats over-sampling." Workshop on learning from imbalanced datasets II. Vol. 11. Washington DC: Citeseer, 2003.
Reviewer 2 Report
The work proposes deep learning based methodology to estimate the attributes of pedestrians as it would be common in application as surveillance or image monitoring. The methodology is able to be implemented in different application such as self-driving vehicles to surveillance. The work is very well organized and written. The methodology is clearly explained and the same goes to the experiments, results and conclusions. Minor comments below:
- Figures 7, 12 and 13 should be improved to be more clear (size of the fonts, legend, number of series in the plot, etc)
Round 2
Reviewer 1 Report
The font size can be further increased to improve the visualization quality in Figure.9 and Figure.7.
For Figure.13, I would suggest the authors add X (Epoch) and Y (loss) labels to improve the image quality.
